

# First complete mitogenomes of Diamesinae, Orthocladiinae, Prodiamesinae, Tanypodinae (Diptera: Chironomidae) and their implication in phylogenetics

Chen-Guang Zheng[1], Xiu-Xiu Zhu[1], Li-Ping Yan[2], Yuan Yao[3], Wen-Jun Bu[1], Xin-Hua Wang[1] and Xiao-Long Lin[1]

[1] College of Life Sciences, Nankai University, Tianjin, China
[2] School of Ecology and Nature Conservation, Beijing Forestry University, Beijing, China
[3] College of Life Sciences, Tianjin Normal University, Tianjin, China

Corresponding author
Xiao-Long Lin, lin880224@gmail.com

## ABSTRACT

**Background**. The mitochondrial genome (mitogenome) has been extensively used for phylogenetic and evolutionary analysis in Diptera, but the study of mitogenome is still scarce in the family Chironomidae.

**Methods**. Here, the first complete mitochondrial genomes of four Chironomid species representing Diamesinae, Orthocladiinae, Prodiamesinae and Tanypodinae are presented. Coupled with published mitogenomes of two, a comparative mitochondrial genomic analysis between six subfamilies of Chironomidae was carried out.

**Results**. Mitogenomes of Chironomidae are conserved in structure, each contains 37 typical genes and a control region, and all genes arrange the same gene order as the ancestral insect mitogenome. Nucleotide composition is highly biased, the control region displayed the highest A + T content. All protein coding genes are under purifying selection, and the ATP8 evolves at the fastest rate. In addition, the phylogenetic analysis covering six subfamilies within Chironomidae was conducted. The monophyly of Chironomidae is strongly supported. However, the topology of six subfamilies based on mitogenomes in this study is inconsistent with previous morphological and molecular studies. This may be due to the high mutation rate of the mitochondrial genetic markers within Chironomidae. Our results indicate that mitogenomes showed poor signals in phylogenetic reconstructions at the subfamily level of Chironomidae.

# INTRODUCTION

The typical mitochondrial genome (mitogenome) of insects is a double-strand circular molecule ranging from 14kb to 20kb in size, which encodes 37 genes (13 protein-coding genes, two ribosomal RNA genes, and 22 transfer RNA genes) and a control region (*Boore, 1999*; *Cameron, 2014*; *Wolstenholme, 1992*). Due to its small genome size, maternal inheritance, low sequence recombination, and fast evolutionary rates (*Brown, George &*

*Wilson, 1979*; *Curole & Kocher, 1999*), the mitogenome is considered as powerful marker for phylogenetic and evolutionary analysis (*Condamine et al., 2018*; *Jacobsen et al., 2012*; *Stokkan et al., 2018*; *Tang et al., 2019b*). Due to high-throughput sequencing technology, an increasing number of complete mitogenomes have been sequenced among the Diptera, covering most families (*Kang, Li & Yang, 2016*; *Li et al., 2020*; *Miao et al., 2020*; *Ramakodi et al., 2015*; *Tang et al., 2019a*). Mitogenomes have been widely used for mitochondrial structure comparison and phylogenetic analysis at different taxonomic level of the Diptera (*Chen et al., 2018*; *De Oliveira Aragão et al., 2019*; *Miao et al., 2020*; *Yan et al., 2019*; *Zhang et al., 2016*; *Zhang et al., 2019b*). However, complete mitogenomes are still scarce for the family Chironomidae, which limits our understanding of their mitochondrial structure and phylogenetic pattern. In addition, it is still unknown whether mitogenomes can effectively resolve phylogenetic relationships at the subfamily level within Chironomidae.

The dipteran family Chironomidae is a diverse aquatic insect group, and are important bioindicators for freshwater ecosystem monitoring. Within Chironomidae, several phylogenetic studies have been conducted based on morphological characters or combining genetic markers to reconstruct the evolutionary history of subfamilies (*Cranston, Hardy & Morse, 2012*; *Sæther, 1977*), but no one has attempted to use mitogenomes. Prior to this study, only five mitogenomes of Chironomidae were available (*Beckenbach, 2012*; *Deviatiiarov, Kikawada & Gusev, 2017*; *Kim, Kim & Shin, 2016*; *Park et al., 2020*; *Zhang et al., 2019a*), representing species from three subfamilies: Chironominae, Podonominae, and Prodiamesinae. However, comparative analysis of mitogenome structure, base composition, substitution and evolutionary rates among subfamilies has not been carried out. In addition, the monophyly of Chironomidae has not been supported by a recent study using mitogenomes of Culicomorpha (*Zhang et al., 2019b*).

In the present study, we provide complete mitogenomes for four species representing the subfamilies Diamesinae, Orthocladiinae, Prodiamesinae, and Tanypodinae. Along with the published mitogenomes of subfamilies Chironominae and Podonominae, the first comparative analysis of the genome structure, base composition, substitution and evolutionary rates among six chironomid subfamilies is presented. In addition, a phylogenomic analysis covering six chironomid subfamilies was carried out.

## MATERIALS & METHODS

### Taxon sampling

Complete mitogenomes of six chironomid species (Appendix S1), representing six subfamilies, were analyzed in this study, with two ceratopogonid species used as outgroups. The mitogenomes of four non-biting midge species, *Potthastia* sp. (Diamesinae), *Rheocricotopus villiculus* (Orthocladiinae), *Prodiamesa olivacea* (Prodiamesinae) and *Clinotanypus yani* (Tanypodinae) are documented for the first time. The mitogenomes of *Chironomus tepperi* (Chironominae) and *Parochlus steinenii* (Podonominae) were retrieved from GenBank (*Beckenbach, 2012*; *Kim, Kim & Shin, 2016*). The mitogenome of *Propsilocerus akamusi* (MN566452) (*Zhang et al., 2019a*) was excluded from the present study because it is incomplete and lacks annotation. In addition, two species

**Table 1  Taxonomic information, sampling metadata, GenBank accession numbers and references of mitochondrial genomes used in the study.**
.

| Family | Subfamily | Species | Life stage | Sampling metadata | Accession no | Reference |
|---|---|---|---|---|---|---|
| Ceratopogonidae | Ceratopogoninae | *Culicoides arakawae* | | | NC_009809 | *Matsumoto et al. (2009)* |
| Ceratopogonidae | Forcipomyiinae | *Forcipomyia makanensis* | | Makan, Zunyi, Guizhou, China, 27.630765°N, 106.848949°E | MK000395 | *Jiang et al. (2019)* |
| Chironomidae | Chironominae | *Chironomus tepperi* | | | JN861749 | *Beckenbach (2012)* |
| Chironomidae | Diamesinae | *Potthastia* sp. | Adult male | Wuying, Yichun, Heilongjiang, China, 48.0869°N, 129.2470°E, 27-Jui-2016, leg. C. Song | MW373523 | This study |
| Chironomidae | Orthocladiinae | *Rheocricotopus villiculus* | Adult male | Tianmu Mountain National Nature Reserve, Hangzhou, Zhejiang, China, 30.3222°N, 119.442°E, 22-Jul-2019, leg. X.-L. Lin | MW373526 | This study |
| Chironomidae | Podonominae | *Parochlus steinenii* | | King George Island, West, Antarctica, 62.2333°S, 58.7833°W, summer in 2015 | KT003702 | *Kim, Kim & Shin (2016)* |
| Chironomidae | Prodiamesinae | *Prodiamesa olivacea* | Larva | Jiuzhaigou Valley Scenic and Historic Interest Area, Sichuan, China, 33.1928°N, 103.8942°E, 12-Jul-2019, leg. X.-Y. Ge | MW373525 | This study |
| Chironomidae | Tanypodinae | *Clinotanypus yani* | Adult male | Jiulongshan Nature Reserve, Guangyuan, Sichuan, China, 31.976379°N, 106.035644°E, 8-Aug-2017, leg. C. Song | MW373524 | This study |

of Ceratopogonidae (*Culicoides arakawae* and *Forcipomyia makanensis*) (*Jiang et al., 2019*; *Matsumoto et al., 2009*) were selected as outgroups for phylogenetic analyses since Ceratopogonidae was strongly supported as the sister group of Chironomidae in previous studies (*Kutty et al., 2018*). Detailed taxon sampling information is listed in Table 1. The vouchers of the newly sequenced species are deposited at the college of Life Sciences, Nankai University, Tianjin, China.

## DNA extraction, sequencing and assembling

For the newly sequenced species, total genomic DNA was extracted from the body, (except abdomen and genitalia) using a General AllGen Kit (Qiagen, Germany). The entire mitogenome of each species were sequenced using the Illumina NovaSeq 6000 platform

with an insert size of 350-bp and a paired-end 150-bp sequencing strategy by the Allwegene Company and Novogene Co., Ltd. at Beijing, China. About 2 Gb clean data were obtained from each library after trimming using Trimmomatic (*Bolger, Lohse & Usadel, 2014*).

To ensure the accuracy of the mitogenome sequences, three frequently used assembly methods were applied to each sample. A *de novo* assembly was performed using IDBA-UD (*Peng et al., 2012*) with minimum and maximum k values of 40 and 120 bp, respectively. Two reference based assemblies were performed using Geneious 2020.2.1 (*Kearse et al., 2012*) with default setting and MITObim 1.9 (*Hahn, Bachmann & Chevreux, 2013*). The mitogenome sequences obtained by the three methods were aligned, manually compared, and finally compiled into a single sequence in Geneious 2020.2.1 (*Kearse et al., 2012*).

## Genome annotation, composition and substitution rate

Genome annotation was conducted as previously described in *Zheng et al. (2020)*. Specifically, the transfer RNA (tRNA) genes and their secondary structures were detected by MITOS2 webserver (http://mitos2.bioinf.uni-leipzig.de/index.py) (*Bernt et al., 2013*) with invertebrate mitochondrial genetic code. The ribosomal RNA (rRNA) genes were predicted by alignment with homologous regions of mitogenome from closely related species. Protein coding genes (PCGs) were initially annotated using the Open Reading Frame Finder (ORF Finder) as implemented at the NCBI website (https://www.ncbi.nlm.nih.gov/orffinder/) and then compared with published mitogenomes of insects using the program BLAST (http://blast.ncbi.nlm.nih.gov/Blast.cgi). Newly sequenced mitogenomes were submitted to GenBank (accession numbers: MW373523–MW373526).

CGView Server V 1.0 (*Grant & Stothard, 2008*) was used to draw mitogenome maps. Codon usage of PCGs and nucleotide composition were calculated in MEGA X (*Kumar et al., 2018*). The bias of the nucleotide composition was measured according to the formulas: AT-skew = $(A-T)/(A+T)$ and GC-skew = $(G-C)/(G+C)$. Rates of non-synonymous substitution rate (Ka) and synonymous substitution rate (Ks) were calculated in DnaSP 6.12.03 (*Rozas et al., 2017*).

## Phylogenetic analyses

Phylogenetic analyses were conducted using the sequences of 13 PCGs and two rRNAs. The PCGs were aligned based on amino acid sequences using Muscle implemented in MEGA X (*Kumar et al., 2018*). The rRNAs were aligned using MAFFT 7.402 (*Katoh & Standley, 2013*) with algorithm G-INS-i strategy. Alignments of individual genes were then concatenated using SequenceMatrix v1.7.8 (*Vaidya, Lohman & Meier, 2011*) to generate five datasets: PCG123 (all three codon positions of the 13 PCGs), PCG123R (all three codon positions of the 13 PCGs and two rRNAs), PCG12 (the first and second codon positions of the 13 PCGs), PCG12R (the first and second codon positions of the 13 PCGs and two rRNAs), and AA (amino acid sequences of the 13 PCGs). To test substitution saturation, transition and transversion rates were evaluated by DAMBE 5.6.14 (*Xia, 2013*). The program PartitionFinder 2.0 (*Lanfear et al., 2017*) was used to infer the best substitution model (Table S1). The analysis of Bayesian inference (BI) and maximum likelihood (ML) were conducted for each dataset. The BI analyses were performed under

the program MrBayes 3.2.7a (*Ronquist et al., 2012*) with partitioned models (Table S1). Two simultaneous Markov chain Monte Carlo (MCMC) runs of 10,000,000 generations were conducted, trees were sampled every 1000 generations with a burn-in of 25%. The program Tracer 1.7 (*Rambaut et al., 2018*) was used to assess the convergence of runs. The ML analyses were conducted using the program RAxML 8.0.12 (*Stamatakis, 2014*) under the substitution model GTR +GAMMA +I. The nodal support values were calculated with 1,000 bootstrap replicates.

## RESULTS

### Mitogenome organization and composition

The complete mitogenomes of *Chironomus tepperi*, *Potthastia* sp., *Rheocricotopus villiculus*, *Parochlus steinenii*, *Prodiamesa olivacea*, and *Clinotanypus yani* are 15,652, 15,913, 15,985, 16,803, 16,190, and 16,247 bp in size, respectively (Fig. 1; Appendix S2). They are circular molecules, each containing 37 typical mitochondrial genes (13 PCGs, two rRNAs, and 22 tRNAs) and one control region. Among these genes, four PCGs (ND1, ND4, ND4L, and ND5), eight tRNAs (trnC, trnF, trnH, trnL (UAG), trnP, trnQ, trnV, and trnY), and two rRNAs (12S rRNAs and 16S rRNAs) are encoded by the minority strand (N strand), while the other 23 genes are located in the majority strand (J strand). ATP8-ATP6 and ND4L-ND4 overlap by seven nucleotides (ATGATAA and ATGTTAA, respectively) in all six Chironomidae species.

Nucleotide composition (Table 2) of the six Chironomidae species is similar, with a high A+T bias (72.4%–76.8%), the control region has the highest A+T content while the first and the second codon positions of PCGs have the lowest A+T content. All six Chironomidae species exhibited negative AT-skew and GC-skew. All three codon positions of PCGs had negative AT-skew, the GC-skew of the first codon position was positive, while the 2nd and the 3rd codon position were negative. Some gene regions exhibited different nucleotide skew among the six Chironomidae species. For example, in 12S rRNA, the AT-skew of *Chironomus tepperi* and *Clinotanypus yani* are −0.01 and 0.00 respectively, while the AT-skew are positive (0.01–0.05) in the remaining four species.

### Protein coding genes

Among Chironomidae species, most PCGs initiate with the standard start codon ATN. The start codon of COI was TTG in *Chironomus tepperi*, *Potthastia* sp., *Rheocricotopus villiculus* and *Prodiamesa olivacea*. The start codon of ND5 in *Chironomus tepperi*, *Potthastia* sp., *Rheocricotopus villiculus*, *Prodiamesa olivacea* and *Clinotanypus yani* was GTG. ND1 started with TTG in *Potthastia* sp., *Rheocricotopus villiculus*, *Parochlus steinenii*, *Prodiamesa olivacea*, and *Clinotanypus yani*. Most PCGs have complete termination codons (TAA or TAG), however, COII in *Parochlus steinenii* and *Clinotanypus yani* has an incomplete termination codon (T-).

Total codon number (except the termination codons) in *Chironomus tepperi*, *Potthastia* sp., *Rheocricotopus villiculus*, *Parochlus steinenii*, *Prodiamesa olivacea*, and *Clinotanypus yani* were 3,730, 3,743, 3,726, 3,729, 3,729, and 3,709, respectively. The most frequently codon families are Ile, Leu2, and Phe (>300), while the least used codon family is Cys (<50)
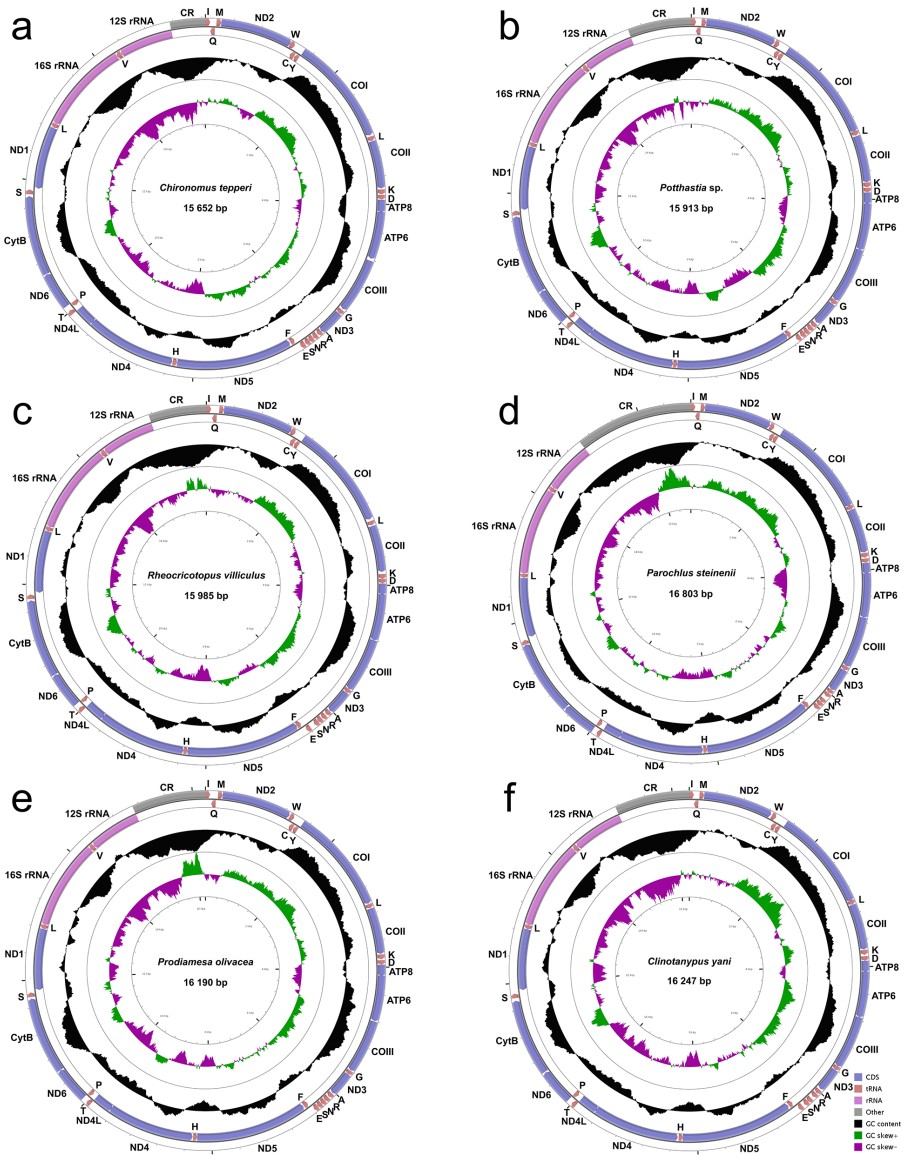

**Figure 1** **Mitogenome maps of *Chironomus tepperi* (A), *Potthastia* sp. (B), *Rheocricotopus villiculus* (C), *Rheocricotopus villiculus* (D), *Prodiamesa olivacea* (E), *Clinotanypus yani* (F).** The names of PCGs and rRNAs are indicated by standard abbreviations, while names of tRNAs are represented by a single letter abbreviation. The first circle shows the gene map and arrows indicate the orientation of gene transcription. Blue arrows refer to PCGs, purple arrows refer to rRNAs, red arrows refer to tRNAs and grey arrow refers to control region. The second circle shows the GC content, which is plotted as the deviation from the average GC content of the entire sequence. The third circle shows the GC-skew, which is plotted as the deviation from the average GC-skew of the entire sequence. The innermost circle shows the sequence length.

in all six Chironomidae species (Fig. 2). The relative synonymous codon usage (RSCU) patterns among the six Chironomidae species are similar. The RSCU values are showed in Fig. 3. All synonymous codons of 20 amino acids are present. The most frequent used codons are NNU and NNA for each amino acid (Fig. 3).

**Table 2   Nucleotide composition of mitochondrial genomes of the six Chironomidae species.**

| | Species | Whole genome | Protein coding genes | First codon position | Second codon position | Third codon position | tRNA genes | 12S rRNA | 16S rRNA | Control region |
|---|---|---|---|---|---|---|---|---|---|---|
| A+T% | *Chironomus tepperi* | 76.7 | 74.3 | 67.6 | 67.6 | 87.6 | 79.0 | 82.6 | 84.3 | 93.0 |
| | *Potthastia* sp. | 76.8 | 74.7 | 69.0 | 66.1 | 88.9 | 76.8 | 78.1 | 82.7 | 93.3 |
| | *Rheocricotopus villiculus* | 77.3 | 74.4 | 69.6 | 67.4 | 86.0 | 79.5 | 84.1 | 84.4 | 93.7 |
| | *Parochlus steinenii* | 72.4 | 69.0 | 64.6 | 64.7 | 77.5 | 73.2 | 76.4 | 80.1 | 85.5 |
| | *Prodiamesa olivacea* | 75.8 | 73.4 | 66.7 | 65.5 | 88.2 | 76.2 | 78.1 | 81.9 | 89.2 |
| | *Clinotanypus yani* | 75.0 | 72.5 | 65.4 | 65.1 | 87.0 | 75.7 | 79.1 | 81.3 | 88.7 |
| AT-Skew | *Chironomus tepperi* | −0.14 | −0.20 | −0.09 | −0.41 | −0.13 | 0.03 | −0.01 | 0.00 | −0.11 |
| | *Potthastia* sp. | −0.13 | −0.20 | −0.10 | −0.39 | −0.12 | 0.03 | 0.01 | 0.04 | −0.05 |
| | *Rheocricotopus villiculus* | −0.12 | −0.18 | −0.09 | −0.40 | −0.08 | 0.02 | 0.01 | 0.05 | −0.07 |
| | *Parochlus steinenii* | −0.11 | −0.19 | −0.09 | −0.40 | −0.11 | 0.04 | 0.05 | 0.03 | 0.06 |
| | *Prodiamesa olivacea* | −0.12 | −0.19 | −0.10 | −0.40 | −0.09 | 0.03 | 0.04 | 0.01 | 0.02 |
| | *Clinotanypus yani* | −0.13 | −0.19 | −0.10 | −0.39 | −0.10 | 0.02 | 0.00 | 0.03 | −0.08 |
| GC-Skew | *Chironomus tepperi* | −0.06 | −0.02 | 0.19 | −0.18 | −0.12 | −0.13 | −0.37 | −0.36 | −0.43 |
| | *Potthastia* sp. | −0.03 | 0.02 | 0.27 | −0.16 | −0.15 | −0.12 | −0.24 | −0.28 | −0.31 |
| | *Rheocricotopus villiculus* | −0.04 | −0.01 | 0.25 | −0.18 | −0.17 | −0.09 | −0.23 | −0.34 | −0.19 |
| | *Parochlus steinenii* | −0.04 | −0.01 | 0.21 | −0.17 | −0.10 | −0.06 | −0.21 | −0.26 | −0.18 |
| | *Prodiamesa olivacea* | −0.04 | 0.00 | 0.25 | −0.16 | −0.21 | −0.09 | −0.24 | −0.29 | −0.16 |
| | *Clinotanypus yani* | −0.06 | 0.00 | 0.24 | −0.18 | −0.18 | −0.12 | −0.28 | −0.34 | −0.39 |

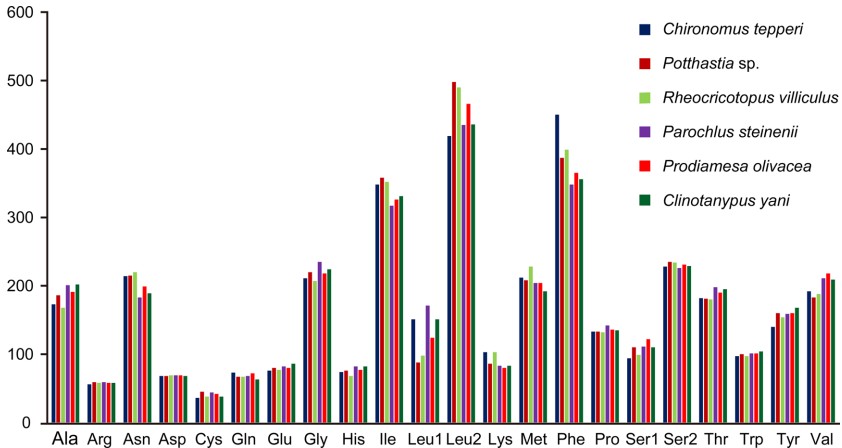

**Figure 2   Patterns of codon usage of the six mitogenomes of six chironomid subfamilies.** The *X*-axis shows the codon families and the *Y*-axis shows the total codons.

The Ka/Ks value ($\omega$) was used to test for signatures of natural selection (*Cheng et al., 2018*; *Hu & Banzhaf, 2008*). The $\omega$ value of all PCGs are less than 0.6. Among the 13 PCGs, ATP8 has the largest $\omega$ value, indicating that ATP8 evolves at the fastest rate. The animal DNA barcoding gene COI has the lowest $\omega$ value (Fig. 4).

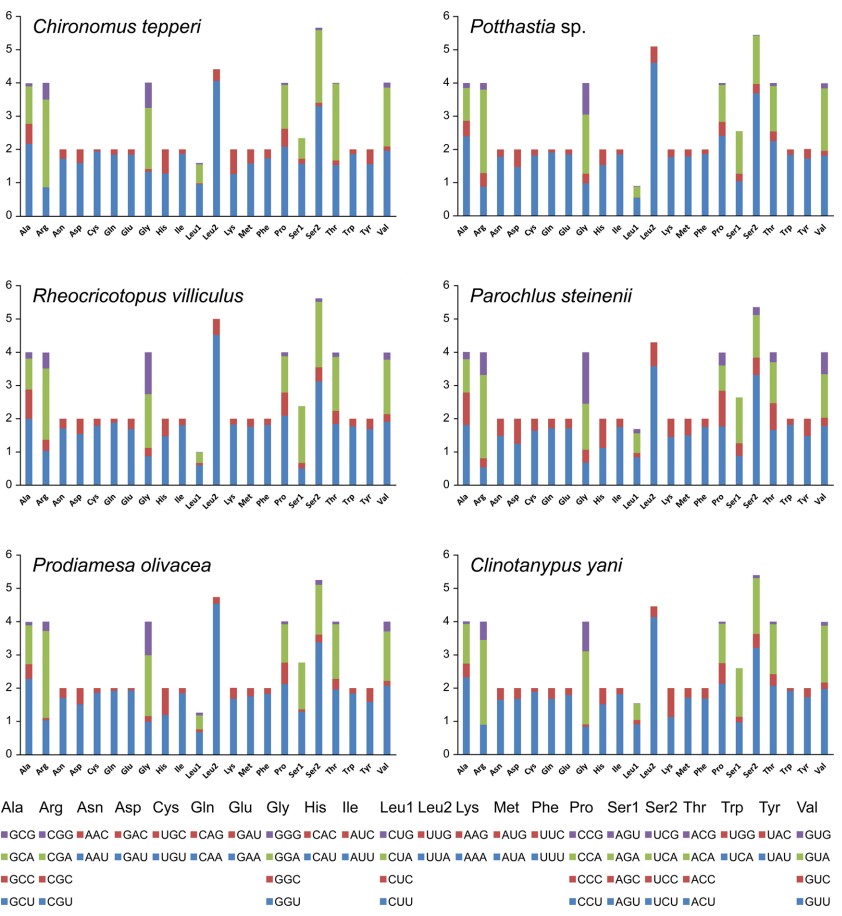

**Figure 3** **The relative synonymous codon usage (RSCU) in the six mitogenomes of six chironomid subfamilies.** The *X*-axis shows the codons and the *Y*-axis shows RSCU values.

## tRNAs, rRNAs and control region

The typical set of 22 tRNA genes were identified in the mitogenomes of all six Chironomidae species, ranging from 63 to 72 bp in length. The tRNA genes exhibit high A+T content (73.2%–79.5%), positive AT-skew, and negative GC-skew (Table 2). All the predicted tRNAs can be folded into the typical clover-leaf secondary structure except trnS (GCU), which lacks the dihydrouridine (DHU) arm. The non-Watson-crick base pair G-U is common in tRNA genes from all Chironomidae species (Fig. S1–S6).

Both 12S and 16S rRNA genes exhibit similar position and size across the Chironomidae mitogenomes. The A+T content of 12S and 16S rRNA genes ranges from 76.4% to 82.6% and 80.1% to 84.4%, respectively. Both genes exhibit positive AT-skew and negative GC-skew in all Chironomidae species except *Chironomus tepperi*: the AT-skew of 12S rRNA and 16S rRNA in *Chironomus tepperi* is −0.01 and 0.00, respectively (Table 2).

The control regions of *Chironomus tepperi*, *Potthastia* sp., *Rheocricotopus villiculus*, *Parochlus steinenii*, *Prodiamesa olivacea*, and *Clinotanypus yani* are 500, 911, 832, 1,783,

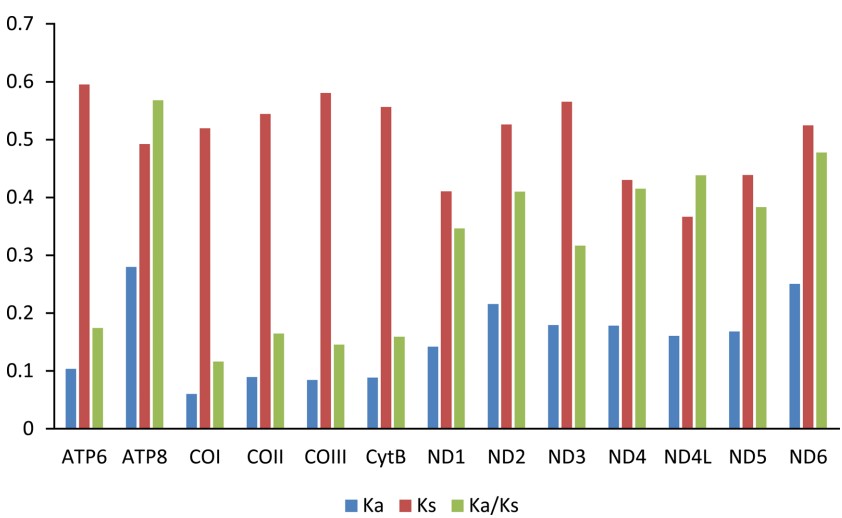

**Figure 4** **Evolution rate of each PCG of the six mitogenomes of six chironomid subfamilies mitogenomes.** Ka refers to non-synonymous substitution rate, Ks refers to synonymous substitution rate, Ka/Ks refers to evolution rate of each PCG.

1,079, and 1,095 bp in size, respectively (Appendix S2). All are A + T rich (85.5%–93.7%), much higher than the whole mitogenomes (72.4%–77.3%).

### Saturation test and phylogenetic analyses

Saturation tests were performed for the four nucleotide datasets. Each dataset was free of saturation (Fig. S7). In general, phylogenetic trees support the monophyly of the Chironomidae across different datasets in ML and BI analyses (Fig. 5, PP = 1, BS = 100). Within Chironomidae, four topologies were inferred from five datasets: (i) Orthocladiinae + (Chironominae + ((Diamesinae + Prodiamesinae) + (Podonominae + Tanypodinae))) was inferred from the PCG123 and PCGR datasets (Figs. 5A and 5B); (ii) Orthocladiinae + (Chironominae + (Diamesinae+ (Prodiamesinae + (Podonominae + Tanypodinae)))) was inferred from the PCG12 dataset (Fig. 5C); (iii) (Orthocladiinae + Chironominae) + (Diamesinae + (Prodiamesinae + (Podonominae + Tanypodinae))) was inferred from the PCG12R dataset (Fig. 5D); (iv) Chironominae + (Orthocladiinae + (Prodiamesinae + (Diamesinae + (Podonominae + Tanypodinae)))) was inferred from the AA dataset (Fig. 5E). The topology inferred from the AA had the strongest nodal support. Based on five different datasets, Podonominae is sister to Tanypodinae with strong support in both BI (PP ≥ 0.98) and, ML (BS = 100) reconstructions, which makes the sister to (Diamesinae + Prodiamesinae) with strong support (PP = 1, BS ≥ 91) at the "tip" position. The remaining subfamilies Chironominae and Orthocladiinae are sister to above four subfamilies.

## DISCUSSION

### Mitogenome features

The entire mitogenome length of the six Chironomidae species differs considerably (15,652–16,803 bp), mainly due to the variation in control region size. All Chironomidae

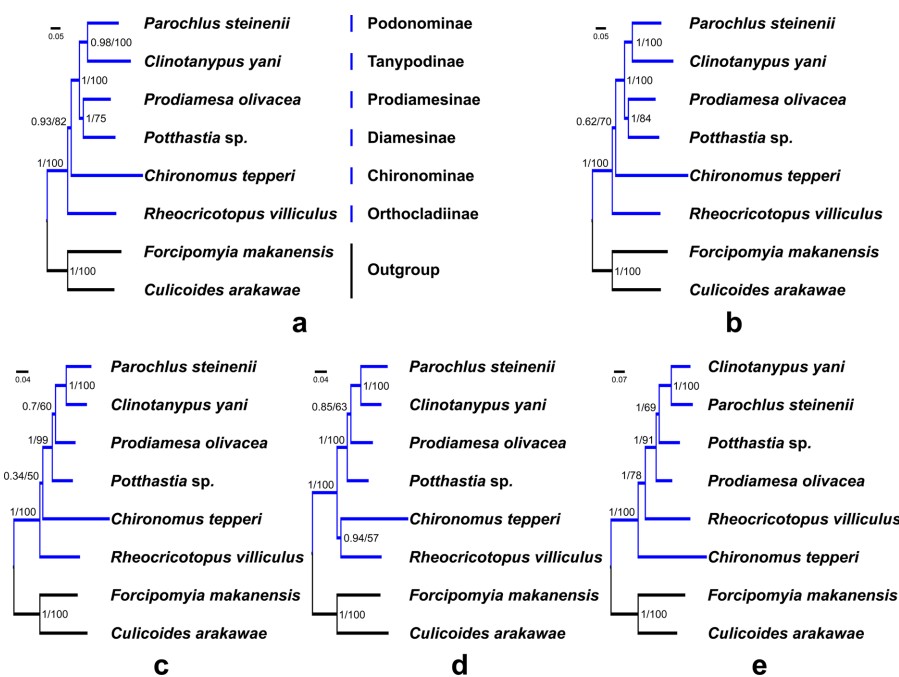

**Figure 5** **Phylogenetic relationships of six subfamilies within Chironomidae inferred from mitogenomes.** (A) Topology obtained based on PCG123; (B) Topology obtained based on PCG123R; (C) Topology obtained based on PCG12; (D) Topology obtained based on PCG12R; (E) Topology obtained based on AA. Numbers at the nodes are BI posterior probabilities (left) and ML bootstrap values (right).

mitogenomes contain 37 typical genes and a control region, the order and arrangement of these genes are completely accordant with the ancestral insect gene arrangement (*Clary & Wolstenholme, 1985*). The whole mitogenome of Chironomidae has high A+T content and similar AT/GC-skew, consistent with the similar base composition biases of insect mitochondrial DNA (*Wei et al., 2010*). This type of nucleotide bias may be related to the asymmetric mutation processes during replication (*Hassanin, Leger & Deutsch, 2005*).

Among the Chironomidae mitogenomes, most PCGs have complete termination codons, while the COII gene in *Parochlus steinenii* and *Clinotanypus yani* has an incomplete termination codon (T-) that probably completed by post-transcriptional polyadenylation (*Ojala, Montoya & Attardi, 1981*). The patterns of codon usage among the Chironomidae mitogenomes are nearly the same. The most frequent used codons were NNU and NNA for each amino acid, reflecting the AT bias of nucleotide composition. For most amino acids, the most frequently used codon is not the anti-codon that strictly correspond to tRNA. The low $\omega$ value for each PCG indicates that they are all under strong purifying selection. The animal DNA barcoding gene COI has the lowest evolutionary rate, which is consistent with the results observed from other insect groups (*Li et al., 2020*; *Yang, Yu & Du, 2013*; *Zhang & Ye, 2017*).

All six Chironomidae mitogenomes contain the 22 typical tRNA genes, and secondary structure across species is similar. Unlike other tRNA genes, trnS (GCU) lacks the dihydrouridine (DHU) arm. This could be commonly found in published insect

mitogenomes (*Li et al., 2012*; *Lu, Huang & Deng, 2020*; *Zhang et al., 2018*). The A+T contents of 12S rRNA, 16S rRNA, and control region are much higher than that in the whole genome in Chironomidae mitogenomes, indicating a strong A+T bias in these regions.

### Phylogenetic analyses

In this study, we applied a variety of strategies to explore the phylogenetic relationships of six subfamilies within the Chironomidae, and confirmed the monophyly of Chironomidae (Fig. 5). However, the topology of subfamilies based on mitogenomes in this study is inconsistent with previous morphological and molecular studies (*Cranston, Hardy & Morse, 2012*; *Sæther, 1977*; *Sæther, 2000*). The present morphological phylogenetics of Chironomidae (*Sæther, 2000*) is composed 11 subfamilies, including (((((Chironominae + Orthocladiinae) + Prodiamesinae) +Diamesinae) + Buchonomyiinae + Chienomyiinae) + ((Usambaromyiinae + Tanypodinae) + Podonominae + Aphroteniinae)) + Telmatogetoninae. The present molecular phylogenetic system of Chironomidae (*Cranston, Hardy & Morse, 2012*) is composed nine subfamilies, including ((((((Chironominae + (Orthocladiinae + Prodiamesinae)) + Diamesinae) + Telmatogetoninae) + Tanypodinae) + Podonominae) + Aphroteniinae) + Buchonomyiinae. Nevertheless, Podonominae and Tanypodinae are ancestral taxa based on both traditional morphological and molecular phylogenies. However, they appear at the "tip" position of mitogenomic phylogenetic tree. Moreover, the "tip" taxa Chironominae and Orthocladiinae appear at the "root" position of the mitogenomic phylogenetic tree. This erroneous phylogenetic reconstruction may be a result of long branch attraction (LBA) (*Siddall & Whiting, 1999*). Due to the high mutation rate of the mitochondrial genetic markers within Chironomidae, some studies (*Ekrem & Willassen, 2004*; *Ekrem, Willassen & Stur, 2010*) have reported that mitochondrial markers (e.g., COI, COII) are not suitable for phylogenetic relationship reconstruction. Here, our mt data reveal different evolutionary history of six subfamilies, which is contradictory with traditional morphology-based systematics. Therefore, we assume that mitogenomes has poor signal for phylogenetic reconstructions at subfamily level in the Chironomidae.

## CONCLUSIONS

In this study, we sequenced four complete mitogenomes representing four subfamilies of Chironomidae by whole genome sequencing technologies and did the first comparative analysis of mitogenome base composition and evolutionary history in Chironomidae. The study shows that mitogenomes of Chironomidae are conserved in structure, gene order and nucleotide composition. Our results revealed that mitogenomes have poor phylogenetic signals for subfamily level relationships in Chironomidae.

## ACKNOWLEDGEMENTS

A big thank to Dr. Lidong Mo for his help on the manuscript improvement. We also thank Dr. Andrey Krasheninnikov and another two anonymous reviewers for their constructive comments.

### Funding

This research was funded by the National Natural Science Foundation of China, grant number 31900344, and China Postdoctoral Science Foundation Grant, grant number 2018M640227. The funders had no role in study design, data collection and analysis, decision to publish, or preparation of the manuscript.

### Grant Disclosures

The following grant information was disclosed by the authors:
National Natural Science Foundation of China: 31900344.
China Postdoctoral Science Foundation Grant: 2018M640227.

### Competing Interests

The authors declare there are no competing interests.

### Author Contributions

- Chen-Guang Zheng analyzed the data, prepared figures and/or tables, and approved the final draft.
- Xiu-Xiu Zhu performed the experiments, analyzed the data, prepared figures and/or tables, and approved the final draft.
- Li-Ping Yan, Wen-Jun Bu and Xin-Hua Wang conceived and designed the experiments, authored or reviewed drafts of the paper, and approved the final draft.
- Yuan Yao performed the experiments, prepared figures and/or tables, and approved the final draft.
- Xiao-Long Lin conceived and designed the experiments, analyzed the data, authored or reviewed drafts of the paper, and approved the final draft.

### Data Availability

The newly sequenced four mitochondrial genomes are available at NCBI SRA (BioProject ID: PRJNA685615), and the assembled sequences are available at GenBank (MW373523–MW373526).

In this study, we also used four published mitochondrial genomes from GenBank: NC_009809, MK000395, JN861749, KT003702

In addition, all mitochondrial genome sequences used in this study are available in the Supplementary File.

### Supplemental Information

Supplemental information for this article can be found online at http://dx.doi.org/10.7717/peerj.11294#supplemental-information.

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
