# Peer review of "First complete mitogenomes of Diamesinae, Orthocladiinae, Prodiamesinae, Tanypodinae (Diptera: Chironomidae) and their implication in phylogenetics"

_PeerJ, doi:10.7717/peerj.11294_

## Round 0.1 · original submission · Major Revisions

Dear Dr. Zheng and colleagues:

Thanks for submitting your manuscript to PeerJ. I have now received three independent reviews of your work, and as you will see, the reviewers raised some concerns about the research. Despite this, these reviewers are optimistic about your work and the potential impact it will have on research studying chironomid phylogenetics and evolution. Thus, I encourage you to revise your manuscript, accordingly, taking into account all of the concerns raised by all three reviewers.

Importantly, please ensure that an English expert has edited your revised manuscript for content and clarity.

Please also ensure that your figures and tables contain all of the information that is necessary to support your findings and observations. Revise incorrect information. The Materials and Methods appear to be missing important information (e.g., specimen metadata, voucher deposit info, etc.). All statistical methods should be adequately described such that they are repeatable.

The interpretations of the phylogeny estimation need to be expanded upon.

Please note that Reviewer 2 kindly provided a marked-up version of your manuscript.

I agree with the concerns of the reviewers, and thus feel that their suggestions should be adequately addressed before moving forward.

I look forward to seeing your revision, and thanks again for submitting your work to PeerJ.

Good luck with your revision,

-joe

·

Basic reporting

no comment

Experimental design

no comment

Validity of the findings

no comment

Additional comments

I commend the authors for their extensive and detailed work. But there is a weakness, it is in the section "Taxon sampling" which should be improved upon before Acceptance.
There is no information about the places of collection of specimens for genetic analysis, dates, collectors and stage of development, the stored place of vouchers.

Reviewer 2 ·

Basic reporting

Written expression isn't up to publishable quality, although the basic layout of the study and their ideas are sufficient to edit it into shape. References could be used more extensively, but more or less OK. Figures are good.

I've attached a marked up copy of the text with editorial suggestions.

Experimental design

Pretty basic experimental design for sequencing. Comparative and phylogenetic analysis is better designed. Methods are acceptable and fairly well documented.

Validity of the findings

Comparative genomics findings are well enough done, even if they basically find nothing particularly novel across the 'subfamilies within a family' scope of the study. The phylogenetic results seem pretty robust on the face of it but are pretty thoughtlessly dismissed as erroneous as they don't confirm previous studies. I think that the authors need to drill more on this and explore those previous findings before they can make the conclusion that they arrive at.

Additional comments

Zheng et al. present a useful expansion of our knowledge of chironomid mitochondrial genomes. The sequencing and comparative genomics portions of the paper are well done, even if not particularly novel - it is still useful to know that the genome is fairly conservative across these taxonomic ranges. The phylogenetic analysis is similarly well done but the write up and discussion of it is poor and needs more work before the final conclusions of the paper can be accepted. The writing overall needs considerable improvement and a marked up copy of the manuscript is attached with editorial suggestions for improving the text.

1) Citations. The paper could probably do with more citations (e.g. for the published mt genomes used) and I've marked some places in the text. But more generally think about if you are giving enough credit to those who contributed to the thoughts expressed in the paper.

2) Phylogeny. The description in the results is really poorly done, sorry to say. It is imprecise about relationships and jumps between levels in the tree which make is hard to convey meaning. You also seemingly report only the outcome of the two inference approaches for one dataset (2 support values) despite doing 5 datasets in the paper (10 support values). You don't even mention how the datasets differ in their topologies (4 support one, 1 support another topology) or whether any of them is more or less likely to be true.

In the discussion, then you more or less completely dismiss your phylogenetic result because it doesn't match that of previous papers (principally the Cranston 2012 paper). You need to rigorously examine the findings in both your own and the prior studies before coming to a conclusion as to which is right. You have pretty good nodal support for relationships between subfamilies (except for made the node Chironominae + (Pod+Tany+Prod+Diam)) - how well supported are subfamily nodes in Cranston 2012? Do either of the Saether papers include discussion of morphological features which would be consistent with any of your nodes. It is common for morphology phylogenetic papers to present both a preferred topology and a discussion of contrary evidence - that contrary evidence might support your relationships.

Rather than just give up, decide that mitogenomes can't do subfamily relationships and call it all long branch attraction evaluate how good those previous studies really were in light of these data.
By the by, data sensitivity analysis such as you've done is usually taken as a test for long branch attraction, so that the trees are pretty insensitive to it suggests long-branch effects are not that big an issue here. A more in depth consideration of the phylogenetic results is needed prior to publication.

Annotated reviews are not available for download in order to protect the identity of reviewers who chose to remain anonymous.

Reviewer 3 ·

Basic reporting

The English writing should be improved before submission. Please pay particular attention to English grammar and sentence structure.

Experimental design

No comment.

Validity of the findings

No comment.

Additional comments

Below are some corrections and comments that I believe could help improve the manuscript:

Lines 196-197: The three last numbers (803, 805, 801) do not agree with the sizes of the D-loop of the respective species presented in Appendix S2. Please correct.
Line 197: The A+T content of the analyzed sequences is comparable to the A+T content of the D-loop of most insects. Perhaps the authors could consider modifying the "remarkably" expression.
Line 205: Please correct "families" to "sub-families".
Lines 207-211: Phylogenetic trees in Figure 5 do not all present the exact same topology. Perhaps the authors could consider describing the topologies in more detail.
Lines 215-216: I believe that the difference of about 1200bp among mitogenomes is considerable. I would suggest modifying the "slightly" expression. The fact that this difference is due to the size variation of the control region is rather interesting. Maybe the authors could comment on that (is there any sequence similarity of D-loop among species? Could size differences be attributed to repeats in the D-loops of larger sizes?)
Table 1: The table does not contain references. The authors should either add this information in the table or modify the Table title accordingly.

---

## Round 0.2 · Minor Revisions

Dear Dr. Zheng and colleagues:

Thanks for revising your manuscript. The reviewers are very satisfied with your revision (as am I). Great! However, there are a few minor edits to make. Please address these ASAP so we may move towards acceptance of your work.

Best,

-joe

·

Basic reporting

no comment

Experimental design

no comment

Validity of the findings

no comment

Additional comments

I'm asking you to correct "Prof. Andrey Krasheninnikov" to "Dr. Andrey Krasheninnikov" or to remove the phrase altogether

Reviewer 2 ·

Basic reporting

No comment

Experimental design

No comment

Validity of the findings

No comment

Additional comments

Suggestions from prior review have been well incorporated in this revision.

Reviewer 3 ·

Basic reporting

no comment

Experimental design

no comment

Validity of the findings

no comment

Additional comments

In the revised manuscript the authors address all of my comments.

The English language has been seriously improved but a few additional corrections are proposed:
line 39: replace "molecular " with "molecule"
line 64: "...by in a recent..." delete "in"
line 67: replace "Allowing " with "Along"
lines 158-159: "...while positive AT-skew (0.01–0.05) in the remaining four species." The verb is missing from this sentence.
line 164: Please replace "ND1 was TTG" with "ND1 started with TTG"
line 232: "...from in other..." delete "in"
line 265: "we sequenced the four..." delete "the"
line 266: Replace "...technologies. And did..." with "...technologies and did..."

---

## Round 0.3 · accepted · Accept

Dear Dr. Zheng and colleagues:

Thanks for revising your manuscript based on the concerns raised by the reviewers. I now believe that your manuscript is suitable for publication. Congratulations! I look forward to seeing this work in print, and I anticipate it being an important resource for groups studying chironomid phylogenetics and evolution. Thanks again for choosing PeerJ to publish such important work.

Best,

-joe